# Safety, Tolerability, and Immunogenicity of V160, a Conditionally Replication-Defective Cytomegalovirus Vaccine, in Healthy Japanese Men in a Randomized, Controlled Phase 1 Study

**DOI:** 10.3390/antib12010022

**Published:** 2023-03-10

**Authors:** Shinya Murata, Nobuyuki Oshima, Takashi Iwasa, Yukako Fukao, Miyuki Sawata

**Affiliations:** MSD K.K., Tokyo 102-8667, Japan

**Keywords:** cytomegalovirus, congenital infection, immunogenicity, Japan, neutralizing antibody, pregnancy, V160, vaccine, safety

## Abstract

Cytomegalovirus (CMV) infection can cause newborn morbidity and mortality; no pharmacological method of reducing CMV infection during pregnancy is currently available. In a phase 1 study in the United States, V160, a conditionally replication-defective CMV vaccine, was immunogenic and well tolerated. This placebo-controlled study (NCT03840174) investigated the safety and immunogenicity of a three-dose V160 vaccine administered over six months. A total of 18 healthy adult Japanese males (9 seronegative and 9 seropositive) were enrolled at a single center and randomized 2:1 to intramuscular V160 or placebo. In vitro, V160 induced high CMV-specific neutralizing antibody (NAb) titers (50% neutralization titer [NT_50_], 3651; 95% confidence interval [CI], 1688–7895) in the CMV-seronegative per-protocol immunogenicity (PPI) population one month after the third vaccine dose was administered compared with no change in the placebo arm (NT_50_, <94; 95% CI < 94–115). The geometric mean titer ratio in the seronegative population versus baseline was 77.7 (95% CI, 23.9–252.4). CMV NAb titers in the CMV-seropositive PPI population were similar to baseline NAb titers observed in the CMV-seropositive population. V160 was well tolerated, and no vaccine viral DNA shedding was observed. In conclusion, the immunogenicity and safety profile of V160 in Japanese participants was consistent with other populations.

## 1. Introduction

Cytomegalovirus (CMV) infection is the most frequent cause of newborn malformation in developed countries, resulting in hearing loss, neurological deficits, and developmental delays in up to 20% of infants with congenital infection [1,2]. Approximately 300,000 babies born in Japan each year are at risk of congenital CMV infection [3], and around 1 in 1000 live births in Japan result in congenital CMV infection-related disability, which is similar to the incidence of Down syndrome [3]. Accordingly, the annual economic burden of congenital CMV infection in Japan was estimated to be JPY 27.6 billion in 2019, predominantly due to the social costs associated with congenital infection [4].

Approximately 83% of people worldwide express anti-CMV immunoglobulin G (CMV seropositivity), indicating past infection [5]. In Japan, the reported prevalence of CMV positivity is lower, particularly in people of child-bearing age, at approximately 58% among individuals in their 20s, increasing to approximately 73% among individuals in their 30s [6]. This means that many Japanese women of childbearing age are at risk of CMV infection, especially because adults are most often infected after being exposed to the virus when caring for infected children who are excreting CMV in their urine, saliva, or other secretions [7,8]. CMV can also be transmitted via blood transfusion, breast milk, sexual intercourse, and transplanted organs [2,9].

In addition, CMV infection is generally asymptomatic in healthy individuals [8], so when symptoms of CMV infection occur, such as fever, sore throat, fatigue, and/or swollen glands, they are often mild and easily mistaken for other infectious illnesses [10]. Therefore, maternal CMV infection is infrequently recognized [9]. Furthermore, CMV establishes lifelong latency and may reactivate after infection [8]. Preventive measures, such as educating pregnant women about the risk of CMV infection and using appropriate hygiene measures, are relied on to reduce the risk of infection. Still, overall awareness of CMV among expectant mothers in Japan is low [9,11].

There is currently no effective pharmacological method of reducing the risk of infection during pregnancy, or any recognized intervention that can effectively reduce transmission of CMV from a newly infected pregnant woman to her fetus [9]. The maternal adaptive immune response can be effective in reducing the risk of congenital CMV infection, but women who develop primary CMV infection during pregnancy are at particular risk of placental CMV transmission [2]. However, the high frequency of CMV re-infection with a different strain not recognized by the immune system, or reactivation, means that most cases of congenital CMV infection occur in mothers who are seropositive [2]. Therefore, both CMV-seronegative and -seropositive individuals are considered to be candidates for a CMV vaccine [2].

A CMV vaccination is considered to be a high public health priority, but early attempts to develop a vaccine have failed to achieve sufficient immunogenicity, particularly in women of childbearing age [7]. V160 is a vaccine that comprises a replication-defective CMV that effectively induces neutralizing antibodies and a T-cell-mediated response against wild-type CMV [12]. In particular, V160 expresses the CMV pentameric complex necessary to elicit potent neutralizing antibody (NAb) titers [7].

In a phase 1 study conducted in the United States (U.S.), V160 was generally well tolerated with no serious adverse events (SAEs) observed, and only transient injection site reactions were reported that were mild-to-moderate in severity [13]. NAb titers and T cell responses induced by V160 vaccination in CMV-seronegative individuals were consistent with those observed with natural infection and maintained for at least 18 months [13]. Vaccination with V160 has also been found to be effective against a number of genetically distinct clinical CMV isolates and to protect against viral infection of several different types of human cells in vitro [14]. This study aimed to investigate the immunogenicity and safety of the V160 CMV vaccine, including assessing post-vaccination plasma virus levels, viral DNA shedding, and leakage from the injection site, in a healthy Japanese population of both CMV-seronegative and -seropositive individuals.

## 2. Materials and Methods

### 2.1. Study Design

A phase 1, randomized, double-blind, placebo-controlled, single-center safety and immunogenicity study was conducted in healthy adult Japanese males (V160-003-00). This study evaluated the safety and immunogenicity of a 3-dose regimen of V160 human CMV vaccine (100 units with aluminum phosphate adjuvant [225 μg] per 0.5 mL dose) administered intramuscularly (IM) over 6 months. This dosage was selected based on the results from a previous phase 1 study carried out in the U.S. The study was performed in compliance with the International Conference on Harmonisation (ICH) guidelines, Good Clinical Practice guidelines, local Japanese regulations, and in line with the principles of the Declaration of Helsinki. Approval for type 1 use of a living modified organism was obtained under the Cartagena law prior to initiating the study. Institutional review board approval was also obtained prior to initiating the study. All participants provided written informed consent prior to enrollment. The study was prospectively registered on Clinicaltrials.gov (study identifier: NCT03840174) prior to enrolling the first participant.

### 2.2. Study Population

The study enrolled Japanese males who were 20–64 years of age and judged by the investigators to be healthy after obtaining a medical history and a physical examination. Each participant was serologically confirmed to be CMV-seropositive or CMV-seronegative at visit 1 (screening visit; within 21 days prior to vaccination) and agreed to remain abstinent or use contraception for the duration of the study. Participants were ineligible to participate in the study if they had a history of any allergic reaction to any vaccine component; had a recent (<72 h prior to receipt of study intervention) history of febrile illness (oral temperature ≥ 38 °C or equivalent); were immunocompromised or had been diagnosed as having an immunodeficiency, hematological malignancy, or other autoimmune disease that required immunosuppressive medication; had a condition in which repeated venipuncture or injections posed more than minimal risk for the participant; had a major psychiatric illness; had previously received any CMV vaccine; had any live virus vaccine administered or scheduled to be administered in the period ±4 weeks of receipt of study intervention; had any inactivated vaccine administered or scheduled within the period ±14 days of study intervention; had received any immunosuppressive therapy; or had received any antiviral agent (e.g., letermovir, ganciclovir, valganciclovir, foscarnet, or valacyclovir) with proven or potential activity against CMV within 14 days prior to vaccination, or was likely to receive such an agent within 14 days after vaccination.

### 2.3. Study Procedures

Participants were randomized in a 2:1 ratio with stratification by CMV serostatus (seropositive vs. seronegative) to receive 3 IM injections of V160 or placebo (saline solution) (Figure 1). CMV seropositivity was determined by assessing serum CMV immunoglobulin G levels by enzyme immunoassay at visit 1. Study vaccinations were administered intramuscularly at day 1, month 2 and month 6.

### 2.4. Study Assessments

Serum samples were collected from all participants on visit 2 (day 1, prior to the first vaccination) and visit 8 (month 7, 1 month after the third vaccination) to assess NAb titers. Functional antibodies were measured by in vitro viral NAb assay to assess the ability of vaccine-induced immune sera to inhibit the infection of ARPE-19 cells by the AD169rev strain of CMV expressing a green fluorescent protein (GFP) reporter [15]. NAbs present in test serum prevent the entry of CMV into target cells, and the subsequent expression of the GFP reporter. Serum samples were serially diluted and mixed with an epithelial cell-tropic CMV before being added to cells, which were fixed after 48 h of incubation with the serum/virus mixture and then subsequently scanned using an EnSight imager (PerkinElmer Inc, Waltham, MA, USA). Neutralizing activity is presented as the interpolated dilution corresponding to 50% of the maximum (median of the no-serum control wells) and the minimum (median of the no-virus control wells). The lower limit of quantitation was a reciprocal dilution of serum required to inhibit viral infection by 50% (NT_50_) < 94.

Viral DNA was extracted from plasma, urine, saliva, injection site swab, and adhesive tape swab samples and assayed for the presence of CMV (including V160) by a polymerase chain reaction (PCR) assay to evaluate viral detection in plasma, viral DNA shedding, and injection site leakage. Plasma samples were collected prior to vaccination, at 0 min (immediately following vaccination) and 3 h after the first vaccination on day 1 and on days 3, 7, and 14. Saliva and urine samples were collected prior to each vaccination, on days 3, 7, and 14 after the first vaccination, and 1 month after the third vaccination. Injection site swab samples were collected from the injection site 0 min, 10 min, 20 min, and 30 min after vaccination on day 1. Adhesive tape was placed over the injection site after the injection site swab sample was taken and the area was wiped with alcohol swabs, and the tape was replaced at 10 min intervals up to 30 min after vaccination on day 1. Swab samples were taken from both the inside and outside of the used tape at 10 min, 20 min, and 30 min after vaccination on day 1. If CMV DNA was detected in plasma, urine, or saliva samples, the V160 vaccine virus DNA and nonvaccine virus DNA were distinguished using a separate PCR assay.

All participants were observed for 30 min after each vaccination for any immediate reactions. A vaccine report card was used to document solicited injection site adverse events (AEs) occurring on days 1–5 following dosing, oral temperature, solicited systemic AEs, and concomitant medications; in addition, from days 1–14 after each vaccination dose, any other injection site or systemic AEs were collected. Telephone contact was made to remind participants to complete their vaccination report card 14 days after the second and third vaccinations. Heart rate, respiratory rate, blood pressure, and oral temperature were assessed at screening, day 1, month 2, month 6, and month 7.

### 2.5. Study Endpoints

The primary endpoints were solicited injection site reactions at days 1–5 after each vaccination visit and solicited systemic AEs and vaccine-related SAEs at days 1–14 after each vaccination visit. Solicited injection site reactions included pain/tenderness, erythema/redness, and swelling. Solicited systemic AEs included headache, tiredness, muscle pain, and joint pain. Any temperature ≥ 38.0 °C oral or equivalent on days 1–14 following vaccination was considered an AE (fever). Secondary endpoints included CMV-specific NAb titer and detection of V160 viral DNA in plasma, urine, saliva, injection site swab, and adhesive tape swab. Safety and tolerability of the V160 vaccine were also assessed. AEs were recorded using the Medical Dictionary for Regulatory Activities (MedDRA) version 22.1.

### 2.6. Statistics

Participants were considered to have completed the study if they received all 3 doses of the study vaccine at the time points specified in the study protocol and completed the month 7 study visit. The primary immunogenicity analyses were based on the per-protocol immunogenicity (PPI) population, which comprised randomized participants who received all three vaccinations within the vaccination visit window specified in the protocol and had not deviated from the protocol in ways that could affect the immune response to vaccination. Supportive immunogenicity analyses were conducted using the full analysis set (FAS) population, which consisted of all randomized participants who received ≥1 vaccination and had ≥1 post-randomization evaluable serology result. Safety analyses were performed on the all-participants-as-treated (APaT) population, which included all randomized participants who received ≥1 vaccination.

CMV-specific NAb geometric mean titers (GMTs) 1 month after the third vaccination were analyzed for CMV-seronegative participants and CMV-seropositive participants using an analysis of variance model. NAb GMTs were log transformed prior to analysis and the treatment difference and 95% confidence interval (CI) were estimated. Estimates of treatment difference and corresponding 95% CIs were then back-transformed to determine the GMT ratio and its 95% CI. Analyses were performed using observed data only.

A sample size of 9 CMV-seronegative and 9 CMV-seropositive participants and 2:1 randomization was expected to generate evidence of ≤46% of individuals in each category administered the vaccine experiencing a specific AE with 90% confidence if that specific AE was not observed in this study.

## 3. Results

In total, 18 healthy Japanese males were enrolled at a single center. Nine participants were CMV-seronegative and nine were CMV-seropositive. Six CMV-seronegative and six CMV-seropositive participants were randomly assigned to V160 and the remaining six participants to placebo. The first participant had their first study visit on 8 March 2019, and the last study visit for the last participant occurred on 7 November 2019.

Mean (standard deviation) age among the study population was 36.4 (14.3; range, 20–63) years. Seventeen participants received all three vaccine doses. One participant who was CMV-seronegative and randomized to V160 withdrew from the study and was excluded from the FAS population because no post-randomization evaluable serology result was available. Two participants in the V160 CMV-seronegative group (including the participant who discontinued) were excluded from the PPI population due to not receiving all three vaccinations within the vaccination visit window specified in the protocol (Appendix A).

### 3.1. CMV-Specific NAb Titers 1 Month after Dose 3

V160 induced high CMV-specific NAb titers (NT_50_, 3651; 95% CI, 1688–7895) in the CMV-seronegative PPI population 1 month after the third vaccine dose was administered compared with no change in the placebo arm (NT_50_, <94; 95% CI, <94–115) (Table 1). The GMT ratio was 77.7 (95% CI, 23.9–252.4). Similar trends were observed in the FAS population.

In the CMV-seropositive vaccine and placebo participants for the PPI population administered vaccine or placebo, CMV NAb titers at 1 month after the third vaccination were similar between the two groups (Table 1). CMV-specific NAb geometric mean fold rise (GMFR) at 1 month after the third vaccination was 1.8 in the V160 group and 1.2 in placebo, but this was not considered to represent a clinically relevant difference.

### 3.2. Viral Detection in Plasma

CMV viral DNA was detected in plasma on day 3 after the first dose in all participants randomized to V160 (*n* = 12, 100%). Although V160 viral DNA was detected in all seronegative participants, V160 viral DNA was only observed in 3 (50.0%) seropositive participants (Appendix A). For the remaining 3 seropositive participants, wild-type CMV was detected on day 3 in one participant; a discriminate assay could not be performed for the other 2 seropositive participants because of low sample viral loads. CMV viral DNA was not detected in plasma at any other timepoint, except for the immediate assessment (0 min after vaccination) on day 1 in 1 CMV-seronegative participant administered V160.

### 3.3. Viral DNA Shedding

V160 viral DNA was not detected in the urine or saliva of any participants, although non-V160 CMV DNA was observed in saliva samples from 2 (33.3%) CMV-seropositive participants and in a urine sample from 1 (16.7%) CMV-seropositive participant administered V160 during the study. These events were observed on day 14 after the first V160 dose, at month 2 (prior to second dose administration), and at month 7 (after third dose administration). All events were determined to be wild-type virus (i.e., considered to most likely be reactivation of natural infection and unrelated to V160 administration). No viral DNA shedding was observed in any CMV-seronegative participants administered V160 or any participants administered placebo.

### 3.4. Viral Leakage

Injection site swabs were positive for viral leakage for all participants immediately after administering V160 (0 min), but by 30 min after the first injection, viral leakage was only observed for 4/6 (66.7%) seropositive and 3/6 (50.0%) seronegative participants administered the V160 vaccine. In addition, viral DNA was detected on swabs from the inside of adhesive tape samples for 2 (33.3%) and 3 (50.0%) seropositive participants at 10 min and 20 min after V160 vaccination, respectively, and for 1 (16.7%) seronegative participant at 10 min. Viral DNA was not detected on swabs from the inside of adhesive tape at 30 min or on any swabs taken from the outside of the adhesive tape.

### 3.5. Safety

At least one AE (solicited or unsolicited) was reported by 5 (83.3%) and 6 (100%) V160 recipients in the seropositive and seronegative populations, respectively, compared with 2 (66.7%) and 1 (33.3%) seropositive and seronegative participants administered placebo, respectively (Table 2). A higher number of injection site AEs were observed among participants administered V160 compared with those administered placebo among participants who were both CMV-seropositive and -seronegative, but the overall incidence of non-injection site AEs were similar. All injection site pain events were mild in intensity. All injection site erythema and swelling events were <2.5 cm in diameter.

The only systemic AE observed in more than one seronegative participant administered V160 was fatigue (Table 2). All systemic AEs were mild in intensity. No participant experienced fever (oral temperature ≥ 38.0 °C) during the 14-day post-vaccination period. No AEs leading to discontinuation or SAEs were reported. No participants died during the study.

## 4. Discussion

A vaccine against CMV infection has the potential to fill an urgent public health need by reducing the risk of congenital CMV infection [7]. However, attempts to develop anti-CMV vaccines have often resulted in suboptimal titers of NAbs and have demonstrated only modest immunogenicity against CMV infection in CMV-seronegative women [7].

In this study, the V160 vaccine was found to be generally well tolerated in healthy Japanese males and effective in inducing NAb titers in CMV-seronegative participants. It is notable that the outcome of this study is consistent with the results reported in a larger phase 1 study conducted in the U.S. that enrolled both males and females, supporting the generalizability of outcomes from phase 1 studies of V160 [13]. Furthermore, the Japanese participants in this study were younger (mean age, 36 vs. 44 years), offering a study population that was more representative of patients in an age range where parenthood may be expected [13]. This is particularly important given the higher risk of congenital CMV in younger people in Japan and in the context of ethnic differences in the risk of congenital CMV infection being observed in the past [7]. The NAb titer achieved in CMV-seronegative participants using a V160 dose comprising 100 units and aluminum phosphate adjuvant was also consistent with previous observations [13,16]. However, the vaccine administered in this study was lyophilized, whereas the U.S. study used a frozen preparation, and the assay used to make the assessments in this study has key differences. In particular, the NAb assay used in the earlier phase 1 study utilized a near-infrared dye-tagged immunostaining reagent to detect immediate early proteins expressed in CMV-infected ARPE-19 cells, whereas this study applied a NAb assay that utilized a GFP reporter. In particular, the current assay may return lower NAb values than the earlier assay (unpublished data).

The safety profile in this study was also consistent with previous reports. Injection site pain was the most commonly reported injection site reaction, with swelling and erythema reported in a minority of participants [13]. Reports of headache, fatigue, myalgia, and arthralgia were also reported in this study, but at much lower rates than in the U.S.-based study [13]. Furthermore, the absence of V160 viral DNA in urine and saliva samples from all participants, except temporal detection in plasma, also confirms that the V160 vaccine has a replication-defective design, which is consistent with previous observations [13]. Leakage around the injection site, as observed in this study, may be expected following the administration of a vaccine by injection, but this study indicates that V160 is unlikely to be excreted into the environment following injection in standard clinical practice.

Post-vaccination NAb levels in seronegative participants in this study were consistent with baseline NAb levels among CMV-seropositive individuals in a previous study in the U.S., but were below mean baseline levels observed in CMV-seropositive participants [13]. However, the results in seronegative participants are consistent with observations in a similar population enrolled in a phase 2b study in the U.S. [16]. In the absence of an immunologic correlate of protection for the prevention of maternal–fetal CMV transmission, natural immunity offers a reasonable benchmark for evaluating the efficacy of V160 because immunity to CMV, and an early response to primary CMV infection, can protect against maternal–fetal CMV transmission [7].

The ability to vaccinate women of childbearing age who are CMV-seronegative offers an important intervention for reducing the risk of congenital CMV infection, especially as many women with a primary CMV infection may not be correctly diagnosed and the risk of congenital CMV-related disability is greatest when primary infection occurs during the first trimester of pregnancy [9]. In particular, given that prior natural infection decreases the risk of congenital CMV infection by approximately 70%, vaccination against CMV infection may be expected to substantially reduce the burden of congenital CMV infection [9].

However, prior attempts to administer CMV vaccines to CMV-seronegative women have failed to prevent infection when exposed in a daycare setting, even though vaccination demonstrated efficacy in preventing serious disease in transplant patients [7]. The pentameric complex of proteins present on the surface of CMV particles in the V160 vaccine are a key feature because a rapid response to this complex has been linked to protection against placental transmission in pregnant women [7]. A vaccine also offers a useful alternative to other methods of preventing maternal CMV infection that have been investigated but have failed to demonstrate sufficient efficacy to justify further development, such as administering passive immunity using immune globulin [9].

This study is limited by its small sample size and short duration. However, a previous study suggests that the antibody response to V160 is durable [13]. Further information about the immune response, such as T cell responses induced by the vaccine, would also be valuable. In addition, the Japanese study population may also limit generalizability, although this study is largely complementary to an earlier phase 1 study performed in the U.S. [13].

## 5. Conclusions

In conclusion, the immunogenicity and safety profile of the V160 vaccine in this study is generally consistent with the profiles observed in other populations. Further clinical investigation of V160 for the prevention of CMV infection is required to understand whether vaccination can prevent maternal–fetal CMV transmission and congenital CMV infection.

## Figures and Tables

**Figure 1 antibodies-12-00022-f001:**
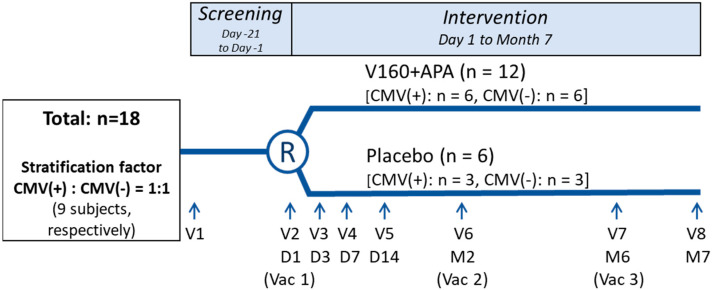
Study design. Abbreviations: APA, aluminum phosphate adjuvant; CMV, cytomegalovirus; D, day; M, month; R, randomization; V, visit; Vac, vaccination.

**Table 1 antibodies-12-00022-t001:** CMV-specific neutralizing antibody GMT (PPI population).

	V160 (*n* = 12)		Placebo (*n* = 6)		GMT Ratio ^a^ (95% CI)
	n	GMT (95% CI)	n	GMT (95% CI)	
**Seronegative**					
Day 1	4	<94 (<94–<94)	3	<94 (<94–<94)	77.7 (23.9–252.4)
1 month after dose 3	4	3651 (1688–7895)	3	<94 (<94–115)	
**Seropositive**					
Day 1	6	6365 (3622–11,184)	3	7975 (3593–17,699)	1.2 (0.8–2.0)
1 month after dose 3	6	11,768 (8851–15,645)	3	9477 (6335–14,177)	

^a^ V160: placebo. Abbreviations: CI, confidence interval; CMV, cytomegalovirus; GMT, geometric mean titer; PPI, per protocol immunogenicity.

**Table 2 antibodies-12-00022-t002:** Solicited and unsolicited post-vaccination AEs ^a^.

	CMV-Seropositive		CMV-Seronegative	
*n* (%)	V160 (*n* = 6)	Placebo (*n* = 3)	V160 (*n* = 6)	Placebo (*n* = 3)
Any AE Injection site Non-injection site	5 (83.3) 1 (16.7)	1 (33.3) 2 (66.7)	6 (100) 2 (33.3)	0 (0) 1 (33.3)
Vaccine-related AEs ^b^ Injection site Non-injection site	5 (83.3) 1 (16.7)	1 (33.3) 1 (33.3)	6 (100) 2 (33.3)	0 (0) 1 (33.3)
Systemic AEs Vaccine-related	1 (16.7) 1 (16.7)	2 (66.7) 1 (33.3)	2 (33.3) 2 (33.3)	1 (33.3) 1 (33.3)
SAEs	0 (0)	0 (0)	0 (0)	0 (0)
AEs leading to discontinuation	0 (0)	0 (0)	0 (0)	0 (0)
**Injection site reactions (Days 1–5) ^c^**				
Injection site erythema	0 (0)	1 (33.3)	3 (50.0)	0 (0)
Injection site pain	5 (83.3)	1 (33.3)	4 (66.7)	0 (0)
Injection site swelling	1 (16.7)	1 (33.3)	2 (33.3)	0 (0)
**Injection site reactions (Days 1–14) ^d^**				
Injection site pain	5 (83.3)	1 (33.3)	5 (83.3)	0 (0)
**Vaccine-related systemic AEs**				
Fatigue	1 (16.7)	1 (33.3)	2 (33.3)	1 (33.3)
Headache	1 (16.7)	0 (0)	1 (16.7)	0 (0)
Arthralgia	0 (0)	0 (0)	1 (16.7)	0 (0)
Myalgia	0 (0)	1 (33.3)	1 (16.7)	0 (0)

^a^ Every subject is counted a single time for each applicable specific injection site AE. ^b^ Determined by the investigator to be vaccine-related. ^c^ Solicited injection site adverse events include pain/tenderness, erythema/redness, and swelling. ^d^ Events excluding injection site swelling and injection site erythema occurring from days 1–5 post-vaccination. Abbreviations: AE, adverse event; SAE, serious adverse event.

## Data Availability

The data sharing policy of Merck Sharp & Dohme LLC, a subsidiary of Merck & Co., Inc., Rahway, NJ, USA, including restrictions, is available at http://engagezone.msd.com/ds_documentation.php accessed on 3 January 2023. Requests for access to the clinical study data can be submitted through the EngageZone site or via email to dataaccess@merck.com.

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
