# Peer review of "Safety, Tolerability, and Immunogenicity of V160, a Conditionally Replication-Defective Cytomegalovirus Vaccine, in Healthy Japanese Men in a Randomized, Controlled Phase 1 Study"

_2073-4468, 2023, doi:10.3390/antib12010022_

Round 1

Reviewer 1 Report

Murata et.al, reported the CMV vaccine, V160, it’s safety and immunogenicity in Japanese men. In this clinical trial, V160 induced neutralizing antibodies to this conditionally replication defective CMV vaccine by humoral immune response.  Overall, the study design, methodologies and assays chosen are sound and the conclusions made are well-founded with the presented results. However, there are MANY PLACES in the manuscript in which the presentation must be improved as noted below

Line 86: What is the basis for selecting only100 units dose and what is the concentration of CMV in 100 units of V160?

Line 127:  NAb titers were measured in serum samples were collected on D1 and 7th month. Why there were no mid time points to measure the NAb to see the pattern of titer although blood was collected on day 3 for detection of viral DNA in plasma?

Line 134: Elaborate epithelial cell-tropic CMV

Lie 142: What primers (sequence) were used in PCR?

Line 216: Remove “<” 94-115.

Line 214:  Does these NAb titres are enough to neutralize the CMV invitro?

Line 219: Is there any significant difference in NAb titres between CMV-seropositive vaccine and placebo group?

Line 270: Correct “Table 2 legend “

Line 307: Does GFP reporter assay is more efficient than near infrared dye-tagged immunostaining?

Line 308: NAb levels in seronegative population in this study are higher than earlier study (Adler et.al, ref. 13)

Supplementary Table 1: Eloborate m/n in table 1B.  

Why only men were chosen in this study?

Reviewer 2 Report

In this study safety, tolerability and immunogenicity of V160 cytomegalovirus vaccine were evaluated in eighteen Japanese males.

This is a Phase I, randomized, double blind, placebo-controlled study. Although the small number of subjects enrolled, the paper supports and extends the work of Adler et al. (JID 2019) to a different population. However it does not add substantially new information.

The response to the vaccine was evaluated in both CMV seropositive and CMV seronegative subjects. The vaccine induced CMV-specific neutralizing antibodies and no vaccine viral DNA shedding was observed. The vaccine was well tolerated and safety.

The authors discuss the possibility to further investigate vaccine immune response by evaluation of CMV-specific cellular response.

Minor revision

-       The title of Table 2 is the same of Table 1

-       Supplementary Table 1: (A) The column n/T is not clear: the number in the brackets is the total number of partecipants? (B) What does it mean m/n?

-       Reference 16, line 425, “cytomegalovirus”
